# Strategic Approach to Aberrant Hepatic Arterial Anatomy during Laparoscopic Pancreaticoduodenectomy: Technique with Video

**DOI:** 10.3390/jcm12051965

**Published:** 2023-03-01

**Authors:** Jiaguo Wang, Jie Xu, Kai Lei, Ke You, Zuojin Liu

**Affiliations:** Department of Hepatobiliary Surgery, Second Affiliated Hospital of Chongqing Medical University, 76 Linjiang Road, Yuzhong District, Chongqing 400010, China

**Keywords:** aberrant hepatic arterial anatomy, laparoscopic pancreatoduodenectomy, “artery-first” approaches, multi-detector computed tomography

## Abstract

Background: It is critical for every pancreatic surgeon to determine how to protect the aberrant hepatic artery intraoperatively in order to safely implement laparoscopic pancreatoduodenectomy (LPD). “Artery-first” approaches to LPD are ideal procedures in selected patients with pancreatic head tumors. Here, we described our surgical procedure and experience of aberrant hepatic arterial anatomy-LPD (AHAA-LPD) in a retrospective case series. In this study, we also sought to confirm the implications of the combined SMA-first approach on the perioperative and oncologic outcomes of AHAA-LPD. Methods: From January 2021 to April 2022, the authors completed a total of 106 LPDs, of which 24 patients underwent AHAA-LPD. We evaluated the courses of the hepatic artery via preoperative multi-detector computed tomography (MDCT) and classified several meaningful AHAAs. The clinical data of 106 patients who underwent AHAA-LPD and standard LPD were retrospectively analyzed. We compared the technical and oncological outcomes of the combined SMA-first approach, AHAA-LPD, and the concurrent standard LPD. Results: All the operations were successful. The combined SMA-first approaches were used by the authors to manage 24 resectable AHAA-LPD patients. The mean age of the patients was 58.1 ± 12.1 years; the mean operation time was 362 ± 60.43 min (325–510 min); blood loss was 256 ± 55.72 mL (210–350 mL); the postoperation ALT and AST were 235 ± 25.65 IU/L (184–276 IU/L) and 180 ± 34.43 IU/L (133–245 IU/L); the median postoperative length of stay was 17 days (13.0–26.0 days); the R0 resection rate was 100%. There were no cases of open conversion. The pathology showed free surgical margins. The mean number of dissected lymph nodes was 18 ± 3.5 (14–25); the number of tumor-free margins was 3.43 ± 0.78 mm (2.7–4.3 mm). There were no Clavien–Dindo III–IV classifications or C-grade pancreatic fistulas. The number of lymph node resections was greater in the AHAA-LPD group (18 vs. 15, *p* < 0.001). Surgical variables (OT) or postoperative complications (POPF, DGE, BL, and PH) showed no significant statistical differences in both groups. Conclusions: In performing AHAA-LPD, the combined SMA-first approach for the periadventitial dissection of the distinct aberrant hepatic artery to avoid hepatic artery injury is feasible and safe when performed by a team experienced in minimally invasive pancreatic surgery. The safety and efficacy of this technique need to be confirmed in large-scale-sized, multicenter, prospective randomized controlled studies in the future.

## 1. Introduction

The origin and course of the hepatic artery outside the liver vary greatly, and there are many types of hepatic artery variation. Many patients have one or more variations in the hepatic artery [1]. In laparoscopic pancreatoduodenectomy (LPD), variations in the celiac trunk (CT) and superior mesenteric artery (SMA) need to be paid special attention to by surgeons, while AHAA is the most common and potentially lethal variation. The occurrence of liver infarction and liver abscess after LPD is mostly related to hepatic arterial vascular injury, which is a rare complication but is an important cause of death after LPD [2,3,4]. The recognition and appropriate management of AHAA is critical during LPD because complications of injury include hepatic ischemia and bilioenteric anastomosis breakdown [5,6]. Previous studies have mainly reported the effect of the aberrant hepatic artery on perioperative outcomes in open pancreatic surgery. Only a few single-center retrospective studies have analyzed the role of AHAA in minimally invasive pancreatic surgery [7,8,9,10,11]. However, we lack a standardized process to safely and effectively protect AHAA during LPD.

In 2006, Pessaux et al., first reported a method in which the SMA is dissected first by severing the posterior pancreatic capsule early during the operation [12]. With technical advancements in surgery for pancreatic cancer, many pancreatic surgeons now recommend a periadventitial approach to the SMA, aiming to achieve the complete removal of the retropancreatic tissue encircling the SMA to maximize the surgical margin around the SMA and to obtain a total mesopancreas excision (TMpE). Currently, many studies [13,14,15,16] focus on the oncological benefits of different SMA-first approaches. Few reports have analyzed the feasibility, safety, and indications of the SMA-first approach for AHAA during LPD. In the present study, we describe the standard operating procedure for AHAA-LPD in our institute, and two types of approaches to AHAA are highlighted. We hope to provide new insights regarding the implementation of AHAA-LPD.

## 2. Materials and Methods

A total of 106 patients were recruited consecutively at the Second Affiliated Hospital of Chongqing Medical University from January 2020 to April 2022. In total, 24 patients underwent AHAA-LPD at the authors’ institution. They gave informed consent for operative management. The indication for surgery was the presence of tumors that were presumed to be malignant, including adenocarcinoma of the duodenum, pancreatic ductal adenocarcinoma, ampullary carcinoma, and cholangiocarcinoma. To explore whether our surgical technique was safe and feasible in the performance of AHAA-LPD, 106 patients with LPD in the same period were divided into standard LPD and AHAA-LPD groups to compare the perioperative complications, technical outcomes, and oncological outcomes. Our institution instituted a formal multidisciplinary tumor board for the treatment of new tumors. All new malignancy cases were presented to discuss them and make decisions regarding them. All patients routinely underwent imaging studies, including blood biochemistry, MDCT, and endoscopic ultrasound, during the preoperative evaluations.

We used the preoperative MDCT data for the 3D reconstruction (CAS Hisense Medical Equipment Co., Ltd. Qingdao, China). We classified several AHAAs which required special intraoperative attention into the following categories (Figure 1A–D):Type A: Replaced right hepatic artery originates from the SMA;Type B: Accessory right hepatic artery originates from the SMA;Type C: Common hepatic artery and right hepatic artery originates from the SMA;Type D: Common hepatic artery originates from the SMA.

We illustrate the advantages of the right posterior approach in revealing the root of the AHAA in the following case. The preoperative 3D reconstruction showed that the rRHA was derived from the SMA (Figure 2A). The duodenum and the head of the pancreas were fully dissociated along a wide Kocher incision to expose the inferior vena cava and the left renal vein (Figure 2B). Next, the SMA root was dissected at the angle between the left renal vein and the inferior vena cava (Figure 2C). We dissected the arterial sheath along the course of SMA to expose the origin of the rRHA (Figure 2D). We continued to complete the right-side Heidelberg triangle lymphadenectomy to expose the root of the CT (Figure 2E). The right posterior approach was carried out to achieve good visualization of the root of the rRHA, CT, and SMA (Figure 2F).

We took advantage of the anterior approach when dealing with AHAA. The working end of the harmonic scalpel ran in the same direction as the aberrant hepatic arteries, which could minimize arterial injury during the process of the skeletonization of the aberrant hepatic arteries. We illustrate the advantages of the anterior approach in the full skeletonization of the CHA and RHA in the following case. The preoperative 3D reconstruction showed that the CHA and RHA originate from the SMA (Figure 3A). The pancreatic parenchyma was dissected along the SMA axis (Figure 3B). The supply arteries of the head of the pancreas—IPDA and uncinate artery—were dissected along the SMA, and the splenic vein (SV) was suspended (Figure 3C). The RHA (Figure 3D) and CHA (Figure 3E) were exposed along the SMA axis, and the skeletonization of the RHA and CHA was performed along the artery. The anterior approach was carried out to achieve the full skeletonization of the CHA and RHA (Figure 3F).

The concept of “the combined SMA-first approach” was introduced. We fully combined the two different “SMA-first” approaches. In the first step, we utilized the right posterior approach to expose and suspend the root of the aberrant hepatic artery; then, we performed the full skeletonization of the aberrant hepatic artery via the anterior approach.

This patient was diagnosed with pancreatic head carcinoma before the operation, and three reconstructions suggested that the rRHA originated from the SMA (Figure 4A). The patient was placed in a supine position with her two legs apart. The trocars were routinely arranged according to the “five-hole method.”

First, the entire abdominal cavity was explored, and no abdominal metastasis was seen. The gastrocolic ligament was opened, and the hepatic flexure of the colon was taken down to explore the head of the pancreas and duodenum without making a Kocher’s maneuver. Due to the heavy tissue edema and adhesion, it was difficult to expose the SMV along the lower margin of the pancreatic neck. In the infracolic region, the avascular area of the transverse mesocolon was opened layer by layer to expose the SMV along the initial position of the third portion of the duodenum and the connection of the Treitz ligament. The assistant lifted the duodenum to the upper left, continued to free the duodenum and pancreatic head, and completely exposed the third portion of the duodenum and the rear of the mesentery. The posterior tunnel of the pancreatic head along the space continued to be established. The left renal vein and inferior vena cava were separated and exposed along the left renal vein and inferior vena cava (Figure 4B). The right posterior approach was prepared. Back in the supracolic region, the stomach was separated along the lower 1/3 of the stomach. The liver was suspended with a “V-shaped” suture to expose the hepatic hilum. To make the extended Kechor incision better and expose the right rear of the hepatoduodenal ligament, we gave priority to cholecystectomy. The number 8a lymph node was removed along the upper edge of the pancreas, and rapidly frozen pathology was performed. The GDA was dissected in combination with careful blunt and sharp dissection and was severed 0.5 cm from the CHA (Figure 4C). Peripheral lymph nodes were resected along the CHA PHA LHA axis. The CHA, the PHA, and the LHA were hung with rubber bands and retracted to the left. The anterior peripheral lymphatic connective tissue was opened along the portal vein, the gap between the portal vein and the bile duct was separated, and the bile duct was completely separated. Particular attention was paid to the protection of the rRHA. Then, the Henle’s trunk and the inferior mesenteric vein (IMV) were exposed and disconnected at the right and left edges of the SMV, respectively. The pancreatic parenchyma was dissected at the vertical axis of the SMA (Figure 4D). An extended Kocher maneuver was carried out via the right posterior approach to achieve good visualization of the aorta and the SMA root. The non-working end of the harmonic scalpel continued to be used along the direction of the longitudinal axis of the SMA to open the vascular sheath, and the lymph nodes around the SMA were cleaned until the rRHA was exposed (Figure 4E). In the same way, the non-working end of the harmonic scalpel was used to walk along the longitudinal axis of the rRHA to open the vascular sheath, fully expose the rRHA, and clean the lymph nodes around the artery. Then, an elastic rubber band was used to suspend the rRHA (Figure 4F). The pancreatic head and duodenum were restored to their original anatomical positions, and the lymphatic nerve connective tissue on the surface of the SMA was opened via the anterior approach. The SMA was exposed with particular attention paid to the protection of the rRHA, and lymph and nerve tissues around the SMA continued to be dissected. The Heidelberg triangle region was completely dissected, and the inferior pancreaticoduodenal artery (IPDA) and uncinate process arteries were dissociated (Figure 4G). The “L” hole was opened at the lower edge of the pancreas and left the vascularless area of the SMA, the proximal jejunum was lifted, and the proximal jejunum was severed 15 cm from the duodenojejunal junction. Vascular clips temporarily blocked PV, SMV, and SV. Then, the space between the pancreatic head and the SMV was separated by a combination of blunt and sharp separation along the SMV until the specimen and SMV were completely dissociated. Finally, for the connection between the rRHA and the specimen (Figure 4H), complete tumor resection was carried out along the longitudinal axis of the rRHA with a harmonic scalpel (Figure 4I). Pancreaticoenterostomy, bilioenterostomy, and gastrointestinal anastomosis were performed successively.

### Statistical Analysis

Descriptive statistics were used to evaluate variants. Continuous numerical variables in accordance with a normal distribution were described with the mean value (standard deviation (SD)). Continuous numerical variables that did not conform to a normal distribution were described with the median (interquartile range (IQR)). Counts and percentages were used to summarize categorical variables. Between the two groups, continuous numerical variables were compared using either the independent samples *t*-test or the Mann–Whitney U test, and categorical variables were compared with the chi-square test. SPSS version 22.0 (IBM SPSS, Inc, Chicago, IL, USA) was used for all analyses.

## 3. Results

All 24 AHAA-LPD patients (mean age 58.1 ± 12.1 years) underwent blood biochemistry and imaging examination before the operation. The combined SMA-first approach was used by the authors to manage 24 cases. Clinical data were collected and analyzed (Table 1). Within these cases, the rRHA arose from the SMA (10 cases), the aRHA arose from the SMA (12 cases), the CHA and the aRHA arose from the SMA (1 case), and the CHA arose from the SMA (1 case). The operation was successfully completed in all patients without conversion to open surgery. The postoperative course was uneventful. The pathological diagnoses were as follows: adenocarcinoma of the duodenum, pancreatic duct adenocarcinoma, cholangiocarcinoma, and ampullary carcinoma. Postoperation ALT and AST were 235 ± 25.65 IU/L and 180 ± 34.43 IU/L, respectively. All the patients obtained negative resection margins, and the tumor-free margins were 3.43 ± 0.78 mm. The median postoperative hospital stay was 17 days (13.0–26.0 days). There was no mortality. The definition and grading system of the postoperative pancreatic fistula was defined by the International Study Group of Pancreatic Fistula (ISGPF). Postoperative complications were defined according to the Clavien–Dindo classification. Four patients had grade I complications: four cases of scattered peritoneal encapsulated effusions. Seven patients had grade II complications: two cases of pneumonia, six A-grade pancreatic fistulas, and four B-grade pancreatic fistulas. Due to the low confluence of the left and right hepatobiliary duct, the bilioenteric anastomosis was far away from the hilum, resulting in delayed bile leak in one case. No patients had grade III complications or above. The postoperative adjuvant therapy for all patients was determined by the multidisciplinary tumor board.

The comparison of the combined SMA-first approach AHAA-LPD and the concurrent standard LPD (82 cases) is summarized in Table 2. In the standard LPD group, eight (9.8%) patients were borderline resectable cases, and six (7.3%) cases received preoperative neoadjuvant chemotherapy. In the AHAA-LPD group, all of the cases were resectable, and none received neoadjuvant chemotherapy. The median operative time for the two groups was 338 min and 344 min, respectively. The median blood loss was greater in the standard LPD group and was statistically different (500 mL vs. 250 mL, *p* < 0.001). The rates of overall complications were 53.6% and 45.8%. Even though there was no significant statistical difference, higher incidence rates of POPF (41.4%), DGE (9.8%), BL (4.9%), and PH (9.3%) were observed in the standard LPD group. In terms of the oncological outcome, the number of lymph node resections was greater in the AHAA-LPD group (18 vs. 15, *p* < 0.001), but there was no statistical difference in the R0 resection rates between the two groups.

## 4. Discussion

The prime difficulty in AHAA-LPD is determining how the variant hepatic artery can be protected intraoperatively. AHAA has a complicated course and small branches, which are easily damaged. As intraoperative AHAA injury may cause a series of complications, such as liver infarction, liver abscess, and bilioenteric anastomosis breakdown, the preoperative recognition and intraoperative protection of AHAA are crucial [5,6]. This study confirmed the safety and feasibility of the combined SMA-first approach using perioperative outcomes in patients with AHAA-LPD. The combined SMA-first approach did not affect the short-term outcome of surgery compared with concurrent standard LPD patients. Furthermore, we investigated the perioperative variables, which showed that the combined SMA-first approach did not increase the operation time while increasing the number of lymph node dissections. To the best of our knowledge, this is the first article in which how to safely implement AHAA-LPD while providing a complete standardized surgical procedure is discussed.

In recent years, with the development of MDCT and vascular three-dimensional reconstruction technology, the vascular variation within the surgical scope, especially AHAA, can be more fully understood before surgery. Here, we assessed the most common and noteworthy variants of hepatic artery in LPD via preoperative 3D vascular reconstruction and divided them into four categories. Compared with the traditional hepatic artery classification, our classification is simpler and more suitable for the preoperative evaluation of LPD. Wang S. et al., found that the AHA group included 127 patients (22.05%) in their single-center study, which is basically consistent with the proportion of AHAAs (22.6%) found in our study [10]. In the literature, the overall preoperative AHAA reporting rate seems to be much higher than what we found, ranging from 15 to 40% [17,18,19]. Most studies have investigated the effect of AHAA on perioperative outcomes in OPD patients [20,21], and only a few studies have reported the effect of AHAA on mini-invasive pancreatic surgery [7,8,9,11]. Both Nguyen et al. [9] and Kim et al. [8] reported that AHAA had no negative effect on the postoperative outcomes of robotic PD. There were no statistically significant differences in the intraoperative data or postoperative complications. Some recent single-center retrospective studies have further confirmed that LPD with the preservation of AHAA is safe and feasible; they confirmed that AHAA is not a prognostic factor for postoperative complications via multivariate analysis [7,17]. In terms of intraoperative data and postoperative complications, our results are consistent with the previous studies, except in terms of blood loss. The main reason for this difference in the results regarding blood loss was that the surgeon was more careful in the dissection of the mesopancreas along the SMA. Nevertheless, the combined SMA-first approach did not increase the operation time; the median operation time (360 min vs. 598 min) in our study was significantly reduced compared with that in the research by Giani A. et al. [7] without severe postoperative complications. Although there was no statistical difference in our results, the rates of postoperative complications or the rates of pancreatic fistulas were lower in the AHAA-LPD group. This confirmed the safety and feasibility of the combined SMA-first approach in AHAA-LPD.

In terms of oncological outcomes, we needed to determine whether the AHAA blocks the dissection of the lymphatic nerve tissue in the Heidelberg triangle. In our study, the number of lymph node resections was greater in the AHAA-LPD group (18 vs. 15, *p* < 0.001), which exceeded the benchmark’s cutoff value (16 lymph nodes) [22]. We performed LPD through the SMA-first approach and showed that a more complete dissection of lymphovascular systems in the Heidelberg triangle (Figure 1, Figure 3F and Figure 4I) could be performed using this technique. It was confirmed that the combined SMA-first approach could improve the oncological radicality. In previous reports, the safety of tumor resection and the oncological benefits of the right posterior approach and anterior approach were reported, respectively [12,23,24,25,26,27]. However, only sporadic reports describe the choice of approach when faced with an aberrant hepatic artery for open pancreaticoduodenectomy [28,29,30]. Lupascu C. et al. [28] think the posterior approach is most suitable for patients with aberrant hepatic arteries, which can expose the AHA early. Noie et al. [29] advocate a ventral approach for the dissection of AHAA instead of the traditional dorsal approach. Although these approaches can effectively avoid damage to the variant hepatic artery during OPD, considering the special endoscopic view of laparoscopy, it is difficult to effectively avoid injury to the AHAA with a single approach. The most prominent disadvantages of the anterior approach is that the course of AHAA cannot be predicted, and it is easy to cause damage to the AHAA during the resection of the hepatoduodenal and para-aortic lymph nodes. Secondly, the splenic vein is blocked in front of the surgical field, which is also an obstacle to the operation. The right posterior approach can expose the origin of the SMA and AHAA early, but the running direction of the AHAA is perpendicular to the direction of the working end of the ultrasonic scalpel, which can easily cause injury when dissecting AHAAs. In addition, this approach cannot guarantee a complete en-bloc resection of the hepatoduodenal and para-aortic lymph nodes. We combined the right posterior approach and the anterior approach in AHAA-LPD, which effectively avoided the shortcomings of the two approaches and maximized the advantages of the two approaches.

Another concern is that these variant hepatic arteries can be sacrificed. Before answering this question, we must first consider the relationship between AHAA and tumors. This requires higher-quality preoperative imaging assessment and an intraoperative interim decision. Arterial resection or reconstruction should be considered when the distance between the AHAA and the tumor is less than 1 cm [31]. If this part of the aRHA is relatively small in preoperative imaging evaluation or is accidentally discovered during surgery, if the tumor invades, the sacrifice of the accessory right hepatic artery can be considered. Previous reports [32] have also reported that this is safe and feasible. However, for a relatively thick aRHA or the rRHA, if preoperative imaging clearly shows tumor invasion, arterial resection or reconstruction should be considered [33]. Loos M. et al. [34] demonstrated that the SMA-first approach is effective in locally advanced pancreatic cancer with promising long-term survival. Pancreatectomy with arterial resection and periadventitial dissection after neoadjuvant treatment is safe. For this group of patients with possible hepatic artery injury, we recommend not dissociating the liver as much as possible and fully preserving the collateral circulation of the liver.

Although a small amount of literature has reported that the variant hepatic artery does not affect the safety of LPD, how to avoid its injury more effectively during surgery remains controversial. Although our experience is limited, we believe that the combined SMA-first approach for AHAA-LPD deserves our attention. Understandably, more experience and skills are required to make our program a standard technique.

## 5. Conclusions

In conclusion, during AHAA-LPD, The combined SMA-first approach for peri-adventitial dissection of distinct aberrant hepatic arteries to avoid hepatic artery injury is feasible and safe. Of course, our study has some limitations. The limitations include the retrospective single-center study design, the short observation period, and the heterogeneity of patient selection. The safety and efficacy of this technique need to be confirmed in large-scale-sized, multicenter, prospective randomized controlled studies in the future. All cases in the AHAA-LPD group were resectable cases, and whether this technique is feasible for BRPC and neoadjuvant chemoradiotherapy cases needs to be studied further. The long-term prognosis of this technique for patients undergoing AHAA-LPD will also need to be confirmed in a long-term follow-up in the future.

## Figures and Tables

**Figure 1 jcm-12-01965-f001:**
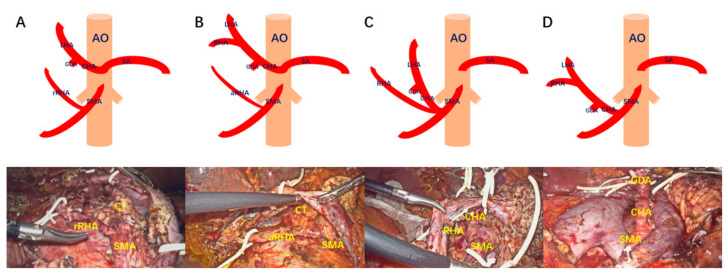
AHAA categorization. (**A**): type A; (**B**): type B; (**C**): type C; (**D**): type D. rRHA: replaced right hepatic artery; aRHA: accessory right hepatic artery; CHA: common hepatic artery; RHA: right hepatic artery; LHA: left hepatic artery; GDA: gastroduodenal artery; SMA: superior mesenteric artery; SA: splenic artery; CT: celiac trunk. Surgical procedure: the right posterior approach to expose the root of AHAA.

**Figure 2 jcm-12-01965-f002:**
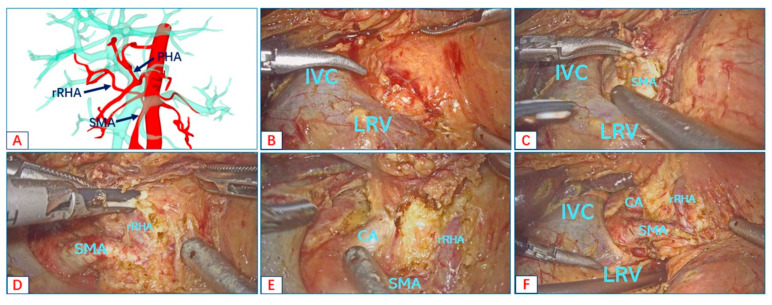
The right posterior approach to expose the root of rRNA. (**A**) The three-dimensional reconstruction of AHAA. (**B**) Exposing the inferior vena cava and the left renal vein along a wide Kocher incision. (**C**) Dissecting the SMA root at the angle between the left renal vein and the inferior vena cava. (**D**) Opening the vascular sheath along the SMA until the rRNA root is exposed. (**E**) The Heidelberg technique for exposing the root of the CT. (**F**) Findings after exposing the root of rRNA. (Additional file 1: https://pan.baidu.com/s/1gIf1wUEs7fMmqw4GWxNkyg?pwd=21ud (accessed on 10 December 2022)). Surgical procedure: the anterior approach is used to fully achieve the skeletonization of aberrant hepatic arterial anatomy.

**Figure 3 jcm-12-01965-f003:**
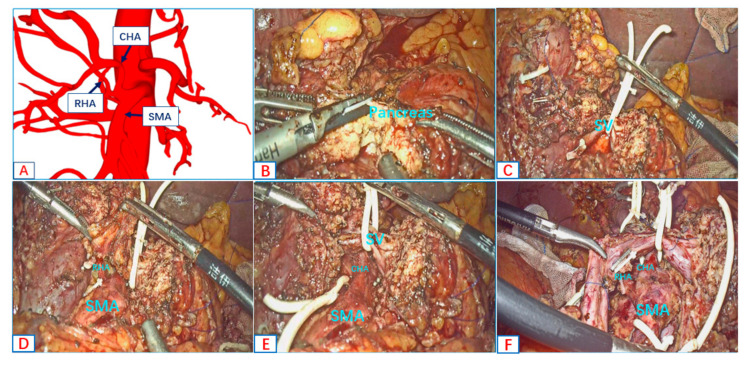
The anterior approach to achieve the full skeletonization of RHA and CHA. (**A**) The three-dimensional reconstruction of AHAA. (**B**) Dissecting pancreatic parenchyma along the SMA axis. (**C**) Dissecting IPDA and uncinate artery along the SMA axis. (**D**,**E**) Skeletonizations of RHA and CHA were performed along the artery. (**F**) Findings after freeing the specimen. (Additional file 2: https://pan.baidu.com/s/1U1bcqS1B7E9c5oDOpktXHg?pwd=wsoa (accessed on 10 December 2022)). Surgical procedure: the combined SMA-first approach for AHAA-LPD.

**Figure 4 jcm-12-01965-f004:**
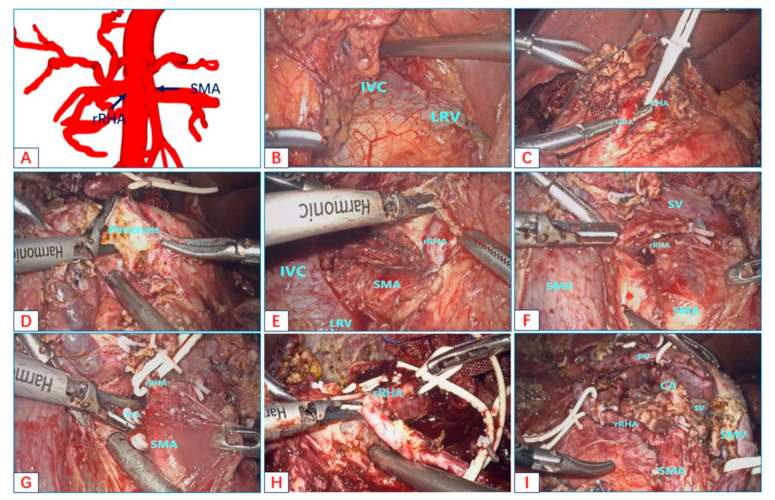
The combined SMA-first approach for AHAA-LPD. (**A**): The three-dimensional reconstruction of the AHAA. (**B**): Exposing the inferior vena cava and the left renal vein along a wide Kocher incision. (**C**): Dissecting the GDA. (**D**): Dissecting pancreatic parenchyma along the SMA axis. (**E**): Exposing and suspending the root of the rRHA by the right posterior approach. (**F**): Dissecting the arterial sheath along the SMA and exposing the root of the rRHA. (**G**): Dissecting the IPDA and uncinate artery along the SMA axis. (**H**): Skeletonization of the rRNA was performed along the arterial sheath via the anterior approach. (**I**): Findings after freeing the specimen. (Additional file 3: https://pan.baidu.com/s/1VVcS4KF-8_WfH_v_o0CawA?pwd=6hf0 (accessed on 10 December 2022)).

**Table 1 jcm-12-01965-t001:** The clinical data of all 24 AHAA-LPD patients.

Clinical Data	Frequencies	Range/Percentage
Sex		
Male	16	66.67%
Female	8	33.33%
Age (year)	58.1 ± 12.1	51–75
Aberrant hepatic arterial anatomy		
rRHA	10	41.67%
aRHA	12	50.00%
CHA and aRHA	1	4.17%
CHA	1	4.17%
Disease		
Pancreatic ductal adenocarcinoma	14	58.33%
Cholangiocarcinoma	4	16.67%
Adenocarcinoma of the duodenum	2	8.33%
Ampullary carcinoma	4	16.67%
Operation time (min)	362 ± 60.43	325–510
Blood loss (mL)	256 ± 55.72	210–350
Conversion to laparotomy	0	0
Postoperation ALT and AST		
ALT (U/L)	235 ± 25.65	184–276
AST (U/L)	180 ± 34.43	133–245
Clavien III–IV complication	0	0
Postoperative pancreatic fistula		
A	6	25.00%
B	4	16.67%
C	0	0
Tumor-free margins (mm)	3.43 ± 0.78	2.7–4.3

rRHA: replaced right hepatic artery; aRHA: accessory right hepatic artery; CHA: common hepatic artery; ALT: alanine aminotransferase; AST: aspartate aminotransferase.

**Table 2 jcm-12-01965-t002:** The comparison of the combined SMA-first approach AHAA-LPD and the concurrent standard LPD.

	Standard LPD (*n* = 82)	AHAA-LPD (*n* = 24)	*p* Value
BRPC	8(9.8%)	0(0%)	0.112
Neoadjuvant chemotherapy	6(7.3%)	0(0%)	0.172
OT	338(280–396)	344(330–388)	0.173
Blood loss	500(375–800)	250(222–281)	<0.001
Overall complications	44(53.6%)	11(45.8%)	0.500
Clavien I–II complication	38(46.3%)	11(45.8%)	0.965
Clavien III–IV complication	4(4.9%)	0(0%)	0.270
30-day mortality	2(2.4%)	0(0%)	0.440
POPF (A, B)	27(32.9%)	10(41.7%)	0.430
POPF (C)	7(8.5%)	0(0%)	0.139
DGE	8(9.8%)	3(12.5%)	0.698
BL	4(4.9%)	2(8.3%)	0.519
PH	6(7.3%)	1(4.2%)	0.585
PHS	16(13–20)	17(14–19)	0.357
Lymph node	15(12–18)	18(16–20)	<0.001
R0	76(92.7%)	24(100%)	0.172

Values are expressed as numbers and percentages or median and IQR. BRPC, borderline resectable pancreatic cancer; OT, operative time; POPF, postoperative pancreatic fistula; DGE, delayed gastric emptying; BL, biliary leakage; PH, postoperative hemorrhage; PHS, postoperative hospital stay.

## Data Availability

The original contributions presented in the study are included in the article. Further inquiries can be directed to the corresponding author.

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
