# Peer review of "Strategic Approach to Aberrant Hepatic Arterial Anatomy during Laparoscopic Pancreaticoduodenectomy: Technique with Video"

_jcm, 2023, doi:10.3390/jcm12051965_

Round 1
Reviewer 1 Report
on line 40, insert the word "potentially" in front of lethal.
please describe the type of software and hardware for doing the arterial reconstructions.
please comment on whether or not using a surgical robot improve the dissection
Author Response
Comment 1: Comments and on line 40, insert the word "potentially" in front of lethal.
Response: Thanks for your comments, we had inserted the word "potentially" in front of lethal.
Comment 2: Suggestions for Authors please describe the type of software and hardware for doing the arterial reconstructions.
Response: Thanks for your comments, we have described the type of software and hardware(CAS Qingdao Hisense Medical Equipment Co., Ltd.) in our manuscript.
Comment 3: Please comment on whether or not using a surgical robot improve the dissection.
Response: Regarding this issue, previous studies have confirmed it. Nguyen et al. [1]first investigated the impact of the aRHAs on the outcomes of minimally invasive PD, considering 142 patients undergoing robotic PD, 31% of whom having RHA anomalies. They found no difference in terms of operative time, estimated blood loss, conversions, postoperative complications, LOS, and readmissions between patients with and without aRHA. These data were successively confirmed by Kim et al[2]. Our center has now started to implement the robotic PD, the da Vinci robot with its unique advantages, robotic surgery is easier to perform than laparoscopic surgery, and it can theoretically improve the dissection. The safety of this technique still requires multicenter, large-sample-sized, prospective, randomized controlled studies to verify.
- Nguyen, T.K., et al., Robotic pancreaticoduodenectomy in the presence of aberrant or anomalous hepatic arterial anatomy: safety and oncologic outcomes.HPB (Oxford), 2015. 17(7): p. 594-9.
- Kim, J.H., et al., Totally replaced right hepatic artery in pancreaticoduodenectomy: is this anatomical condition a contraindication to minimally invasive surgery?HPB (Oxford), 2016. 18(7): p. 580-5.

Reviewer 2 Report
First, I want to congratulate the authors for their work and interest in minimally invasive surgery.
Here are my comments:
1) The authors should mention what is the aim of this retrospective study.
2) The study includes a small number of patients. The authors should mention the limitations of this tudy in drawing the conclusions.
3)The are multiple typographical and English translations errors in the manuscript.
line 38, 46, 56, 92 (Kechor# Kocher), 94, 102,146(stations), 163, 222, 231(Clavien-Dindo classification)
Author Response
Comments and First I want to congratulate the authors for their work and interest in minimally invasive surgery
Response: Thanks for your positive comments.
Suggestions for Authors Here are my comments:
Comment 1: The authors should mention what is the aim of this retrospective study.
Response: Thanks for your comments, We wish to share our team's experiences in implementing the AHAA-LPD and provide our own complete set of standard surgical procedures(with video). We confirmed that the combined SMA-first approach for periadventitial dissection of distinct aberrant hepatic artery avoid hepatic artery injury feasible, and safe. we hope to provide some new insight for the future AHAA-LPD standardized surgery mode. Meanwhile, we present AHAA classification that is more suitable for the preoperative assessment of LPD.
Comment 2: The study includes a small number of patients. The authors should mention the limitations of this study in drawing the conclusions.
Response: Thanks for your comments, we have added the limitations of this study in drawing the conclusions. The limitations include the retrospective single-center study design, the short observation period, and the heterogeneity of patient selection. The safety and efficacy of this technique need to be confirmed by large-scale sized, multicenter, prospective randomized controlled studies in the future. All cases in the AHAA-LPD group were resectable cases, and whether this technique is feasible for BRPC and neoadjuvant chemoradiotherapy cases needs further study. The long-term prognosis of this technique for patients undergoing AHAA-LPD will also need to be confirmed by a long-term follow-up in the future.
Comment 3: The are multiple typographical and English translations errors in the manuscript.
Response: Thanks for your comments, typographical and English translations errors have been corrected and revised one by one in the manuscript.

Reviewer 3 Report
This article describes a retrospective study of 24 patients who underwent laparoscopic pancreaticoduodenectomy in relation to the well-known problem of choosing the most appropriate technical strategy to spare a possible aberrant hepatic artery from the superior mesenteric artery.
There are major concerns I have with this paper.
-I think the authors should organize their paper better by clearly stating the objectives of their retrospective study: is the procedure feasible? Comparison of two possible approaches (posterior or anterior)? Report of the patients outcome? Survival? Complications rate?
-The article is poorly organized as a scientific paper. It seems like a chapter in a surgical textbook describing pancreatic surgical techniques.
-The Authors should highlight what is novel and really interesting in their retrospective study when compared to other dozens of similar articles already published on the well known problem of superior mesenteric artery first approach in pancreaticoduodenectomy and the importance of sparing an aberrant hepatic artery arising from superior mesenteric artery.
- A replaced right hepatic artery should be clearly differentiate unequivocally from an accessory hepatic artery pointing out clearly the anatomical and functional difference and the possible different post-operative risks related to their sacrifice.
- When describing surgical techniques, it is usually believed that the description should be confined to material and methods.
-What are the selection criteria for this cohort of 24 patients? How many patients were borderline resectable patients or oncological resectable patients aiming R0?
-There are too many acronyms; it is very difficult to follow the meaning of all acronyms in this paper. Why the replaced right hepatic artery is rRNA? What this N stand for? It should be more logical rRHA
-Surgical approach description is too long and needs to be shortened.
Some typos; capital and minuscule are missed:
Line 54 Aberrant can be aberrant (aberrant)
Line 56 this article describes the 56 technique of AHAA-LPD. This
Line 94 at the Angle between the left renal, at the angle
Line 97 .the right posterior approach .The
Author Response
This article describes a retrospective study of 24 patients who underwent laparoscopic pancreaticoduodenectomy in relation to the well-known problem of choosing the most appropriate technical strategy to spare a possible aberrant hepatic artery from the superior mesenteric artery.
Response: Many thanks to the reviewers for their approval of the topic of our study.
There are major concerns I have with this paper.
Comment 1: I think the authors should organize their paper better by clearly stating the objectives of their retrospective study: is the procedure feasible? Comparison of two possible approaches (posterior or anterior)? Report of the patients outcome? Survival? Complications rate?
Response: Thanks for your comments, We have revised the content of the manuscript according to the opinions of the reviewers. we add the comparison of the combined SMA-first approaches AHAA-LPD and the concurrent standard LPD. 106 patients with LPD at the same period were divided into standard LPD and AHAA-LPD groups to compare perioperative complications, technical outcome and oncological outcome. We confirmed that the combined SMA-first approach for periadventitial dissection of distinct aberrant hepatic artery avoid hepatic artery injury feasible, and safe.
Comment 2: The article is poorly organized as a scientific paper. It seems like a chapter in a surgical textbook describing pancreatic surgical techniques.
Response: Thanks for your comments, on this issue, we have adjusted the full text according to the opinions of academic editor's comments and all reviewers.
Comment 3: The Authors should highlight what is novel and really interesting in their retrospective study.
Response: Thanks for your comments. Firstly, compared with the Michels[1] and Hiatt[2] classification systems, our proposed AHAA classification is simpler and more suitable for preoperative assessment of LPD. Regarding the novelty of this article, most current studies have mainly investigated the effect of aRHA on perioperative outcomes in PD patients undergoing open surgery. Only a few single-center retrospective studies have analyzed the role of variant hepatic arteries in minimally invasive pancreatic surgery[3-7]. In these studies, no standard surgical procedure was formed in the AHAA-LPA, and it certainly was not reported. As we know, there is no good strategy for the variant hepatic artery in LPD, but more depends on the experience of the surgeon. We first proposed a standardized surgical procedure(with video) for AHAA-LPD, our article is the first to provide a detailed description of the standard surgical procedure (with video) for AHAA-LPD. In our study, all patients in the AHAA-LPD group underwent fixed surgical procedures(the combined SMA-first approach). We confirmed that the combined SMA-first approach for periadventitial dissection of distinct aberrant hepatic artery avoid hepatic artery injury feasible, and safe. According to our surgical procedure, the median operation time(360min vs 598min) in our study was significantly reduced compared with previous studies without severe postoperative complications[3].
Comment 4: when compared to other dozens of similar articles already published on the well-known problem of superior mesenteric artery first approach in pancreaticoduodenectomy and the importance of sparing an aberrant hepatic artery arising from superior mesenteric artery.
Response: the occurrence of liver infarction and liver abscess after LPD is mostly related to hepatic arterial vascular injury, which is a rare complication, but it is an important cause of death after LPD[8-10]. In the long-term clinical follow-up, we found that severe fatty liver may appear in the later stage after hepatic artery injury ( figure). Recognition and appropriate management of AHAA is critical during LPD.
Comment 5: Are placed right hepatic artery should be clearly differentiate unequivocally from an accessor hepatic artery pointing out clearly the anatomical and functional difference and the possible different post-operative risks related to their sacrifice.
Response: The distinction between rRHA and aRHA is well-known problem, rRHA is the only arterial blood supply to the right hemiliver, while aRHA only as part of the blood supply to the the right hemiliver. The sacrifice of aRHA in the presence of large collateral arteries does not cause serious complications[11]. The rRHA should be retained as much as possible, and if tumor invasion or vascular damage exists, arterial resection and reconstruction of the rRHA is required if necessary[12].
Comment 6: When describing surgical techniques usually believed that the description should be confined to material and methods.
Response: Thanks for your comments, the description of surgical techniques Is indeed in the material and methods in our manuscript.
Comment 7: What are the selection criteria for this cohort of 24 patients? How many patients were borderline resectable patients or oncological resectable patients aiming R0?
Response: The inclusion criteria for the AHAA-LPD group were the presence of a variant hepatic artery derived from the superior mesenteric artery and the surgical procedure in accordance with the combined SMA-first approach.
Comment 8: There are too many acronyms, it is very difficult to follow the meaning of all acronyms in this paper Why the replaced right hepatic artery is rRNA? What this N stand for? It should be more logical rRHA.
Response: Thanks for your comments, typographical and English translations errors have been corrected and revised one by one in the manuscript.
Comment 9: Surgical approach description is too long and needs to be shortened.
Response: Thanks for your comments, We did minimize the description of the procedure, but we want to present the complete procedure to better explain the additional video files. We also hope to share some details of our procedure through a detailed description of the surgical procedure.
- Michels, N.A., Newer anatomy of the liver and its variant blood supply and collateral circulation.Am J Surg, 1966. 112(3): p. 337-47.
- Hiatt, J.R., J. Gabbay, and R.W. Busuttil, Surgical anatomy of the hepatic arteries in 1000 cases.Ann Surg, 1994. 220(1): p. 50-2.
- Giani, A., et al., Hepatic vascular anomalies during totally laparoscopic pancreaticoduodenectomy: challenging the challenge.Updates Surg, 2022. 74(2): p. 583-590.
- Kim, J.H., et al., Totally replaced right hepatic artery in pancreaticoduodenectomy: is this anatomical condition a contraindication to minimally invasive surgery?HPB (Oxford), 2016. 18(7): p. 580-5.
- Nguyen, T.K., et al., Robotic pancreaticoduodenectomy in the presence of aberrant or anomalous hepatic arterial anatomy: safety and oncologic outcomes.HPB (Oxford), 2015. 17(7): p. 594-9.
- Wang, S., et al., The Impact of Aberrant Hepatic Artery on Resection Margin and Outcomes of Laparoscopic Pancreatoduodenectomy: A Single-Center Report.World J Surg, 2021. 45(10): p. 3183-3190.
- Zhang, W., et al., A single-center clinical study of hepatic artery variations in laparoscopic pancreaticoduodenectomy: A retrospective analysis of data from 218 cases.Medicine (Baltimore), 2020. 99(21): p. e20403.
- Fujiwara, H., et al., Hepatic infarction following abdominal interventional procedures.Acta Med Okayama, 2004. 58(2): p. 97-106.
- Miura, F., et al., Eleven cases of postoperative hepatic infarction following pancreato-biliary surgery.J Gastrointest Surg, 2010. 14(2): p. 352-8.
- Smith, G.S., B.A. Birnbaum, and J.E. Jacobs, Hepatic infarction secondary to arterial insufficiency in native livers: CT findings in 10 patients.Radiology, 1998. 208(1): p. 223-9.
- Yamamoto, S., et al., Disposal of replaced common hepatic artery coursing within the pancreas during pancreatoduodenectomy: report of a case.Surg Today, 2005. 35(11): p. 984-7.
- Kim, P.T., et al., Aberrant right hepatic artery in pancreaticoduodenectomy for adenocarcinoma: impact on resectability and postoperative outcomes.HPB (Oxford), 2014. 16(3): p. 204-11.

Round 2
Reviewer 3 Report
Please check the English as for Instance: Abstract (Conclusions) and in paragraph 5. Conclusions:
"In conclusions, during AHAA-LPD, the combined SMA-first approach for periadventitial dissection of distinct aberrant hepatic artery avoids hepatic artery injury feasible and safe"
the sentence should be changed in :
"The combined SMA-first approach for peri-adventitial dissection of distinct aberrant hepatic artery to avoid hepatic artery injury is feasible, and safe, when performed by a team experienced with a minimally invasive pancreatic surgery.
Author Response
Dear Editor and Reviewers,
Thank you for offering us an opportunity to improve the quality of our submitted manuscript (Manuscript ID: jcm-2195022). We appreciated very much the reviewers’ constructive and insightful comments. We have submitted the revised manuscript to MDPI’s Author Services for native language revision. Meantime, we have addressed all of these comments. We hope the revised manuscript has now met the publication standard of your journal.
Our point-to-point responses to the queries raised by reviewer 3 are listed.
Comment 1: Please check the English as for Instance: Abstract (Conclusions) and in paragraph 5. Conclusions: "In conclusions, during AHAA-LPD, the combined SMA-first approach for periadventitial dissection of distinct aberrant hepatic artery avoids hepatic artery injury feasible and safe". the sentence should be changed in :
"The combined SMA-first approach for peri-adventitial dissection of distinct aberrant hepatic artery to avoid hepatic artery injury is feasible, and safe, when performed by a team experienced with a minimally invasive pancreatic surgery.
Response: Thanks for your comments for our manuscript. We have made the modifications.
